# Intestinal Flora in Chemotherapy Resistance of Biliary Pancreatic Cancer

**DOI:** 10.3390/biology12081151

**Published:** 2023-08-21

**Authors:** Liuhui Bai, Xiangdong Yan, Jin Lv, Ping Qi, Xiaojing Song, Lei Zhang

**Affiliations:** 1The First Clinical Medical College, Lanzhou University, Lanzhou 730000, China; bailh21@lzu.edu.cn (L.B.); yanxd21@lzu.edu.cn (X.Y.); lvj21@lzu.edu.cn (J.L.); qip21@lzu.edu.cn (P.Q.); songxiaojing4227@126.com (X.S.); 2Department of General Surgery, The First Hospital of Lanzhou University, Lanzhou 730000, China; 3Key Laboratory of Biotherapy and Regenerative Medicine of Gansu Province, The First Hospital of Lanzhou University, Lanzhou 730000, China

**Keywords:** intestinal flora, biliary pancreatic malignancy, chemotherapy resistance, mechanism, regulation

## Abstract

**Simple Summary:**

Chemotherapy is one of the common methods for the treatment of malignant tumors of the biliary pancreatic system, but chemoresistance reduces treatment effectiveness. In recent years, studies have found that intestinal flora imbalance is closely related to chemotherapy resistance. Targeted regulation of intestinal flora can improve chemotherapy resistance. The purpose of this review is to explore the role of intestinal flora in the chemoresistance of malignant tumors of the biliary pancreatic system and to target the regulation of intestinal flora with antibiotics and probiotics so as to seek potential treatment directions.

**Abstract:**

Biliary pancreatic malignancy has an occultic onset, a high degree of malignancy, and a poor prognosis. Most clinical patients miss the opportunity for surgical resection of the tumor. Systemic chemotherapy is still one of the important methods for the treatment of biliary pancreatic malignancies. Many chemotherapy regimens are available, but their efficacy is not satisfactory, and the occurrence of chemotherapy resistance is a major reason leading to poor prognosis. With the advancement of studies on intestinal flora, it has been found that intestinal flora is correlated with and plays an important role in chemotherapy resistance. The application of probiotics and other ways to regulate intestinal flora can improve this problem. This paper aims to review and analyze the research progress of intestinal flora in the chemotherapy resistance of biliary pancreatic malignancies to provide new ideas for treatment.

## 1. Introduction

Malignant tumors of the biliary pancreatic system are highly malignant; most patients are diagnosed at an advanced stage, so the incidence and mortality are close. Cholangiocarcinoma accounts for about 3% of gastrointestinal cancers. The global incidence ranges from 0.3 to 6 cases per 100,000 population per year and mortality from 1 to 6 cases per 100,000 population per year [1]. Pancreatic cancer is the seventh leading cause of cancer-related death worldwide with a global incidence and mortality rate of 4.8 and 4.4 per 100,000 population per year, respectively, and the incidence is increasing year by year [2]. Chemotherapy is the main treatment, and there are several chemotherapy regimens, including FOLFIRINOX (oxaliplatin + irinotecan + leucovorin + 5-fluorouracil), AG (gemcitabine + albumin-bound paclitaxel), GS (gemcitabine + S-1), gemcitabine + cisplatin, gemcitabine + albumin-bound paclitaxel, and other combined treatment regimens, but the efficacy is poor; compared to single-agent chemotherapy, combination chemotherapy can reduce the cytotoxic effect of chemotherapy drugs [3]. Chemotherapy is often interrupted due to recurrent biliary obstruction or inflammation, and drug resistance caused by long-term chemotherapy is not conducive to the treatment of patients, all of which lead to the limitations of chemotherapy [4]. Among them, tumor resistance to chemotherapy drugs is one of the important reasons for the poor clinical efficacy of antitumor therapy observed. Preventing or reducing drug resistance is still a difficult problem to solve to improve the clinical efficacy of chemotherapy [5,6]. In recent years, studies on intestinal flora have found that its imbalance can abolish the protective effect of the intestinal barrier and affect the activity of chemotherapy drugs in vivo metabolically and in other ways, resulting in a reduction in tumor cell sensitivity to chemotherapy drugs and an extension of treatment time, which indicates that the intestinal flora may be closely related to chemotherapy resistance [7,8,9,10,11]. Targeted regulation of intestinal flora may help to reduce chemotherapy resistance and improve the effect of cancer chemotherapy [12,13]. This review aims to interrogate the literature on the role of intestinal flora in the chemotherapy resistance of biliary pancreatic cancer.

## 2. Resistance of Biliary Pancreatic Malignancies to Chemotherapy

Common biliary pancreatic malignancies include gallbladder cancer, cholangiocarcinoma, pancreatic cancer, and pancreatic endocrine tumors. These tumors are characterized by high malignancy, frequent recurrence, and a poor prognosis. Most patients have already missed the opportunity for radical surgery by the time they see a doctor, and chemotherapy is an important treatment method for them [9,14]. The intestinal flora of these cancers is different. The abundance of *Salmonella enterica serovar. typhi* in the stool of patients with gallbladder cancer is increased. Patients with proximal cholangiocarcinoma have a higher abundance of *Helicobacter pylori* and *Escherichia coli* in feces, while patients with distal cholangiocarcinoma have a higher abundance of *Fusobacterium* and Actinobacteria. *Fusobacterium* and *Porphyromonas gingivalis* increase in the feces of patients with pancreatic cancer [15]. In particular, fungi play a unique role in pancreatic ductal adenocarcinoma, which can activate the complement system, participate in the body’s immune response, and promote tumor progression. In addition, the interaction between fungi and bacteria may be involved in the occurrence and development of pancreatic cancer. Although there are few studies on fungi, it may be a new idea for the treatment of pancreatic cancer [16,17].

Tumor resistance is a complex process caused by the change of antitumor drug targets and the decrease in their concentration in cells. It can be divided into inherent and acquired drug resistance. Inherent drug resistance refers to the natural resistance of tumor cells to a certain antitumor drug, which has nothing to do with whether they have been exposed to the drug or not. It may be caused by the expression of mutated oncogenes or tumor suppressor genes in tumor cells, which affect drug resistance. Acquired drug resistance refers to drug resistance of tumor cells being induced by chemotherapy drugs, that is, tumor cells being sensitive to chemotherapy drugs at the beginning of administration and then displaying drug resistance [18,19]. As an important treatment for biliary pancreatic cancer, chemotherapy mainly includes chemotherapeutics such as alkylating agents, antimetabolites, and antibiotics. These drugs can be classified according to different mechanisms of action, as shown in Table 1 [20,21]. The mechanisms of chemotherapy resistance in biliary pancreatic malignancies mainly include increased expression of drug transporters, accelerated drug exclusion, apoptotic dysfunction, tumor stem cell action, and changes in drug metabolism [9]. Studies have shown that the intestinal flora can influence the response of cancer cells to chemotherapy by regulating the local immune response and inflammation around tumors [22]. Intestinal flora can also regulate cancer autophagy through certain signaling pathways, affecting chemotherapy drug resistance [23]. This suggests that intestinal flora may be involved in the process of tumor resistance to chemotherapy drugs.

## 3. Intestinal Flora and Chemotherapy Resistance in Biliary Pancreatic Malignancies

The intestinal flora constitutes a group of bacteria that are designated to plant themselves in the human intestine and depend on the human body for a long time. There are more than 40 bacterial genera and more than 500 bacterial species, mainly composed of obligate anaerobes, facultative anaerobes, and aerobic bacteria, among which obligate anaerobes account for more than 99% and are the dominant bacteria in the intestinal flora, such as *Bifidobacterium*, *Bacteroides*, *Eubacillus*, and *Lactobacillus*. They function in nutrient metabolism and immune regulation. Facultative anaerobic bacteria and aerobic bacteria are mostly pathogenic bacteria, such as *Enterobacter, Enterococcus*, and *Proteus* [7,38]. These bacteria are harmless when the intestinal ecosystem is in balance, but when the flora is disturbed, that is, the type, quantity, and proportion of the normal intestinal flora are changed abnormally from the physiological stoichiometry to pathological ratios, ecological imbalances of the intestinal flora occur, which in turn lead to various diseases and other health problems [39]. Intestinal flora plays an important role in the pathological development of diseases from inflammation to cancer. The increase in pathogenic bacteria such as *Escherichia coli* and *Fusobacterium nucleatum* and the decrease in probiotics such as *Bifidobacterium* and butyrate-producing bacteria lead to the decrease in short-chain fatty acid content, abnormal bile acid metabolism, and DNA damage, which jointly mediate the inflammatory response and induce the progression of diseases to cancer [40,41]. The abuse of antibiotics, radiotherapy, chemotherapy, surgery, trauma, infection, and tumors, as well as environmental degradation, can reduce human immunity, directly damage the intestine, interfere with the physiological mechanism of the host, and cause intestinal flora imbalance, which is characterized by the reduction in probiotics and the increase in pathogenic bacteria, which further leads to various diseases and other health problems [8,42,43]. Environmental factors have a great impact on intestinal flora, especially on dietary structure and the metabolites of intestinal flora (such as fatty acids, choline metabolites, and ethanol metabolites). They regulate the structure of intestinal flora by participating in digestion, nutrient absorption, mucosal immune response formation, and the synthesis or regulation of bioactive compounds and induce changes in the host’s physiological and pathological states [44,45].

In recent years, with the in-depth study of intestinal flora, it has been found that the latter bears a close relationship with chemotherapy drugs. While killing tumor cells, chemotherapy drugs can cause immune imbalance by increasing intestinal permeability, leading to intestinal flora disorders. For example, after the use of oxaliplatin, cisplatin, gemcitabine, capecitabine, 5-fluorouracil, albuminopaclitaxel, irinotecan, and other chemotherapy drugs, the intestinal flora structure of patients changes, increasing the proportion of *Proteus*, *Clostridium*, and other bacteria. Reduced proportions of *Lactobacillus* and *Bacteroides* often result in diarrhea, vomiting, and other adverse reactions [10,46,47,48,49,50,51,52,53]. A decrease in symbiotic bacteria such as *Lactobacillus* and *Bifidobacterium* leads to damage to the intestinal mucosa and changes the structure of the intestinal mucus layer, thus reducing its protective effect on intestinal barrier function [54,55,56,57]. On the contrary, intestinal flora can also affect (enhance or reduce) the efficacy of these chemotherapy drugs and thereby chemotherapy, which may be closely related to chemotherapy resistance [13].

## 4. Mechanism of Intestinal Flora Drug Resistance in Biliary Pancreatic Malignancy Chemotherapy

Cancer chemotherapy resistance results from a complex interplay between gene regulation and the environment. The intestinal flora is involved in the initiation and progression of digestive tract tumors by influencing intestinal inflammation. Pathogenic bacteria in the gastrointestinal tract always cause local inflammation and induce the production of inflammatory cytokines, including interleukin (especially interleukin-1 [IL-1] and interleukin-6 [IL-6]) and tumor necrosis factor-α (TNF-α), and further lead to the activation of tumor-related signaling pathways. The intestinal flora also upregulates Toll-like receptors (TLR) and induces immune tolerance [23,58]. Studies have shown that after chemotherapy, the decrease in intestinal probiotics, such as *Bifidobacterium*, and intestinal mucosal damage lead to the rapid depletion of mucin stored in intestinal goblet cells. The depletion of mucin makes the mucus layer thinner, increases intestinal permeability, and increases the risk of bacterial lipopolysaccharide (LPS) translocation to the circulation where it can easier activate immune cell TLRs. It causes the upregulation of the downstream signaling pathway of the transcription factor nuclear factor kappa B (NF-κB) and the release of proinflammatory cytokines, which reduces the efficacy of chemotherapy drugs [57,59,60].

### 4.1. Gut Microbiota Is Involved in the Mechanism of Gemcitabine Resistance

Studies have found that *Proteobacteria* could metabolize gemcitabine by expressing cytidine deaminase (CDA), changing its chemical structure, and metabolizing the active form of gemcitabine (2′,2′-difluorodeoxycytidine) into inactive 2′,2′-difluorodeoxyuridine, thus influencing its activity and local concentration (Figure 1). Levels of *gamma-Proteus* were also elevated in human pancreatic ductal adenocarcinoma samples compared to normal pancreas samples [13,24,25,26,27,28]. In addition to CDA, pyrimidine nucleoside phosphorylase (PyNPase) produced by *Mycoplasma* can also indirectly enhance the deamination of chemotherapy drugs by removing natural pyrimidine nucleoside, 2′-deoxyuridine, and thymidine, which inhibit gemcitabine deamination, thus adversely affecting the therapeutic effect of chemotherapy drugs (Figure 1) [28,29]. In addition, gemcitabine treatment of pancreatic cancer stimulates the activation of NF-κB and other related inflammatory pathways, increases intestinal permeability, increases *Proteobacteria*, *Escherichia coli*, and *Akkermansia muciniphilia*, and decreases Gram-positive Firmicutes and Gram-negative *Bacteroides*, resulting in reduced butyrate formation, which contributes to its various anticancer properties. In addition, the anti-proliferation, pro-apoptosis, anti-invasion, and anti-angiogenic properties of pancreatic cancer cells cannot be demonstrated, which indicates that gemcitabine chemotherapy brings important changes to the intestinal flora, and this change of intestinal flora will inhibit the anticancer effect of gemcitabine [10,61,62].

### 4.2. Gut Microbiota Is Involved in the Resistance Mechanism of Oxaliplatin and 5-Fluorouracil

Studies in cell lines and mouse models have shown that adhesion proteins (FadA) on *Fusobacterium nucleatum* bind to epithelial cadherin (E-cadherin) to induce β-catenin signaling and regulate inflammatory and carcinogenic responses to promote tumorigenesis [63,64]. Through Toll-like receptor 4 (TLR4) and myeloid differentiation factor 88 (Myd88) of the TLR signaling pathway, *Fusobacterium* induces the selective loss of two autophagy-related microRNAs (microRNA-18a [miR-18a] and microRNA-4802 [miR-4802MYD88]), which activate autophagy, thereby reducing the antitumor activity of oxaliplatin and 5-fluorouracil in cancer patients (Figure 2) [13,23,24,30,65]. The Wnt/β-catenin signaling pathway is the upstream regulatory pathway of some drug resistance proteins, such as the ATP-binding cassette subfamily B member 1 (ABCB1), multidrug resistance 1 (MDR1), and p-glycoprotein (P-gp), which constitute important molecular mechanisms involved in the occurrence, development, metastasis, and chemotherapy resistance of colon cancer and other tumors. MDR1 and P-gp are important members of the ATP-binding cassette effervescent transporter family. Their overexpression has been shown to be one of the most common mechanisms by which hemotherapy occurs. They can cause a large number of anticancer drugs with different structures and functions (such as 5-FU, oxaliplatin, etc.) to be expelled from tumor cells, and they are key effector proteins of chemotherapy resistance [66,67]. Experimental studies have shown that co-culture with *Fusobacterium nucleatum* can enhance the vitality of cancer cells, promote the formation of cell colonies, reduce cell apoptosis, antagonize 5-FU, and enhance chemotherapy resistance and cancer cell proliferation. This is related to the overactivation of the Wnt/β-catenin signaling pathway, which upregulates the ABC transporters (MRP1 and P-gp) (Figure 3) [31]. In addition, immunogenic commensal bacteria (e.g., non-enterotoxin-producing *Bacteroides fragilis*) stimulate follicular helper T cell (TFH) cells to interact with B lymphocytes, enhance oxaliplatin-induced epithelial cell apoptosis, and generate an immune response to enhance the anticancer effect of oxaliplatin [68]. Recent studies have found that butyrate, a metabolite of gut microbiota, can promote the antitumor effect of oxaliplatin through Id2-dependent CD8 T cell immunomodulation. After eliminating such bacteria with antibiotics, cancer cells show increased resistance to oxaliplatin [69].

### 4.3. Gut Microbiota Is Involved in the Mechanism of Irinotecan Resistance

Irinotecan has a heavy piperidine side chain at the C-10 site that can be cut by carboxyl esterase into 7-ethyl-10-hydroxycamptothecin (SN-38), which is 1000 times more potent than irinotecan [70,71]. Both irinotecan and SN-38 are in equilibrium with their active lactone and inactive carboxylate forms, which are pH-dependent. Under acidic conditions, the lactone form is favored, but under physiological or high pH, the lactone form is unstable, and the hydrolysis of the lactone ring into its carboxylate form is favored, at which time the concentrations of *Enterobacterium* and *Proteus* are reduced. This may be related to the activity and resistance of irinotecan [32,72,73,74,75]. The β-glucuronide produced by *Escherichia coli* can convert inactive glucuronide (Sn-38-G) into active SN-38, inhibit β-glucuronide, and reduce irinotecan activity (Figure 4) [30,33,76]. In addition, nuclear factor kappa B (NF-κB) is an anti-apoptotic transcription factor, especially in early transformed tumor cells. Activated NF-κB inhibits the apoptotic cascade induced by tumor necrosis factor α (TNF-α) and chemotherapeutic agents, particularly irinotecan, which are also associated with pathogenic bacteria in the gut [32,34,58,77,78,79,80].

### 4.4. Gut Microbiota Is Involved in the Mechanism of Paclitaxel Resistance

Experimental studies have shown that the imbalance of intestinal flora is closely related to diabetes. The decrease in Firmicutes abundance and the increase in *Proteus* and *Bacteroides* lead to insulin resistance, and high blood sugar increases the therapeutic resistance of pancreatic cancer to citabine/paclitaxel. Additionally, an increase in the hematopoietic stem cell antigen CD133 in the tumor cell population in diabetic models has also been observed. These observations suggest that in animal models of type-2 diabetes, dysregulation of intestinal flora increases resistance to chemotherapy in pancreatic cancer, which may be related to the tumor microenvironment. The specific mechanism needs to be further investigated [35,36,37,81]. Queuosine and S-adenosylmethionine (SAM) are two metabolites of intestinal flora. Queuosine is a rare nucleoside found in tRNA and appears at the swing position of some tRNA anticodons. SAM is an antitumor agent that regulates cysteine–methionine metabolism, immune response, and nucleotide methylation, thereby controlling transcriptional processes. Queuosine enhances the chemoresistance of pancreatic cancer cells to paclitaxel under obesity by protecting cancer cells from chemotherapy-induced oxidative stress by up-regulating the peroxiredoxin1 (PRDX1) recombinant protein. Further experiments showed that this chemoresistance could be reversed by supplementing obese mice with SAM [82].

## 5. Reduction of Chemotherapy Resistance by Regulating the Intestinal Flora

Regulating intestinal flora through antibiotics, probiotics, fecal microbiota transplantation, or nanotechnology may reduce chemotherapy resistance and enhance the antitumor effect of chemotherapy agents (Figure 5). Some mechanisms of tumor drug resistance have been found through existing studies, including high expression of P-gp, overexpression of multidrug resistance-associated protein (MRP), inhibition of apoptosis, etc., but the problem of tumor drug resistance has not been completely solved. In recent years, more and more evidence has shown that intestinal flora plays an important role in inhibiting tumorigenesis and regulating the therapeutic effect of tumors, especially in alleviating the chemotherapy resistance of tumor cells [12,83,84,85]. The imbalance of intestinal flora can seriously affect the pathogenesis and therapeutic effect of cancer. In particular, the modulation of this therapeutic effect is closely linked to the ability of gut microbiota to metabolize antitumor compounds and to modulate the host’s immune response and inflammatory pathways. Together, these two effects could explain the effect of a patient’s gut microbiota pairing on resistance to cancer chemotherapy [86,87].

### 5.1. Antibiotics

Studies have shown that targeted intervention of antibiotics with pathogenic bacteria can enhance the immune function of the body, reduce the metabolism of chemotherapy drugs, and reduce drug resistance so as to restore or even enhance its antitumor efficacy [88]. Although antibiotics can enhance the immunity of the body and improve the sensitivity of tumor cells to some chemotherapy drugs in some ways, the routine use of antibiotics will change the symbiotic microbiota (that is, the bacteria that live with the organism and keep it healthy), interfere with the fixed value of probiotics, and long-term use will lead to the emergence of antibiotic-resistant bacteria strains, resulting in collateral damage to patients [89,90]. Alternatively, antibiotic treatment leads to a decrease in the level of reactive oxygen species produced by the gut microbiota, which is required for the early action of oxaliplatin and cisplatin, which reduces the anticancer efficacy of oxaliplatin and cisplatin. Therefore, the aim of using antibiotics to target intestinal flora to improve the clinical effect of cancer chemotherapy resistance needs more experimental support [91].

### 5.2. Probiotics

Probiotics, which replenish beneficial bacteria that are reduced due to intestinal flora imbalance, may help repair the intestinal barrier, relieve gastrointestinal inflammation, maintain intestinal homeostasis, and reduce adverse reactions, such as diarrhea associated with chemotherapy [92,93]. A randomized, controlled trial has shown that *Bifidobacterium-* and *Lactobacillus*-based probiotics can reduce β-glucuronidase activity, thereby reducing the incidence of diarrhea caused by irinotecan while improving its anticancer activity [94]. Some studies have found that *Lactobacillus paracasei* combined with gemcitabine or 5-fluorouracil in the treatment of pancreatic cancer can reduce the resistance of cancer cells to these two chemotherapy drugs by inducing apoptosis [95,96]. However, some clinical trials have also denied the clinical benefits of probiotics in terms of chemotherapy resistance and efficacy in cancer treatment [97,98,99]. The conflicting results of clinical trials may be explained by interindividual variation in the microbiome and host genome, which may also be related to the limitations of most clinical trials on probiotics (e.g., small sample size, short treatment duration, and lack of follow-up to examine the long-term effects of probiotics on patients).

### 5.3. Fecal Microbiota Transplantation

One study examined the effect of eleven strains of probiotics and autologous fecal microbiota transplantation (aFMT) on mouse and human microbiomes following antibiotic reestablishment. The probiotics significantly delayed microbiome reestablishment. aFMT induced a rapid and nearly complete recovery within days of administration. Based on this finding, aFMT rather than probiotics could be used to reconstruct a patient’s antibiotic-interfered microbiome to reduce tumor resistance in patients [100]. Although fecal microbiota transplantation (FMT) has been shown to be of great therapeutic value in diseases caused by bacteria such as *Clostridium difficile*, it also bears certain risks, such as multidrug resistance, aspiration, and death, which require the use of standard microbial screening in future FMT trials to improve FMT safety [101,102,103].

### 5.4. Nanomaterials

In recent years, new nanotechnologies, such as the clustered regularly interspaced short palindromic repeats (CRISPR)/Cas9 system, provided by phages can target specific bacterial species at the microbiome–cancer interface to minimize interference with the symbiotic microbiome and ensure effective cancer treatment. In a study of colorectal cancer, the covalent attachment of irinotecan-containing glucan nanoparticles to azide-modified phages that inhibit the growth of *Fusobacterium nucleatum* significantly improved the sensitivity of tumor cells to chemotherapy, suggesting that phage-guided nanotechnology can stimulate cancer therapy by regulating the gut microbiome, which may lead to a new approach to cancer therapy [104,105,106]. However, phage therapy to reduce chemotherapy resistance is limited to gastrointestinal tumors, and its effectiveness in the treatment of biliary pancreatic malignancies is still lacking. Nanotechnology can be used to target tumor-associated bacteria or to release anticancer drugs in a controlled manner, thereby increasing the sensitivity of tumor cells to chemotherapy drugs and reducing side effects in patients. Given the impact of nanotechnology on cancer prevention and treatment, efforts should be made to evaluate the mechanisms of nanoparticle-mediated toxicity, side effects, and reduction in chemotherapy resistance with a focus on its role in tumor therapy [107,108,109].

Based on the current research, it is necessary to further clarify the mechanisms of probiotics, FMT, and nanotechnology in antitumor therapy to improve the treatment of biliary and pancreatic malignancies, especially regarding chemotherapy resistance.

## 6. Conclusions and Future Perspectives

To sum up, treatments for biliary pancreatic malignancies are becoming more and more diverse, and combination therapy based on chemotherapy is extremely important. After chemotherapy, along with immune disorders and the destruction of the intestinal epithelial barrier, the types and composition of intestinal flora have undergone significant changes, usually manifested as a reduction in probiotics and an increase in opportunistic pathogens. This pathogenesis may be related to intestinal immune dysfunction after chemotherapy. However, the increase or decrease in some bacteria after an imbalance in the bacterial community has occurred in turn increases or decreases the sensitivity of biliary and pancreatic malignant tumor cells to these commonly used chemotherapy drugs. The changes in intestinal flora can affect the chemoresistance of biliary pancreatic malignancies in a variety of ways, including regulating local immune response and inflammation around the tumor, regulating cancer autophagy through signaling pathways, affecting drug metabolism, increasing the expression of drug transporters, and anti-apoptosis. Such chemotherapy resistance often underlies the poor prognosis and easy recurrence of biliary and pancreatic malignant tumors. By thoroughly investigating the correlation between the intestinal flora and tumor and its treatment, antibiotics, FMT, probiotics, and nano-loaded drug technology have shown their benefits in tumor treatment; they can target the regulation of gut microbiota and improve chemotherapy resistance through the above various ways. And the interaction mechanism between intestinal flora and chemotherapy will also be paid to attract researchers’ future attention. In the future, we need a large number of animal models and clinical trials to further study the role of probiotics and nano-drug loading technology in the chemotherapy of patients with biliary pancreatic malignant tumors so as to fully understand the complex interaction between intestinal flora and chemotherapy resistance in biliary pancreatic malignant tumors. With the improvement of animal experiments and clinical data in the future, the potential of intestinal flora in reducing chemotherapy resistance, improving efficacy, or reducing related adverse reactions will be explored and applied.

## Figures and Tables

**Figure 1 biology-12-01151-f001:**
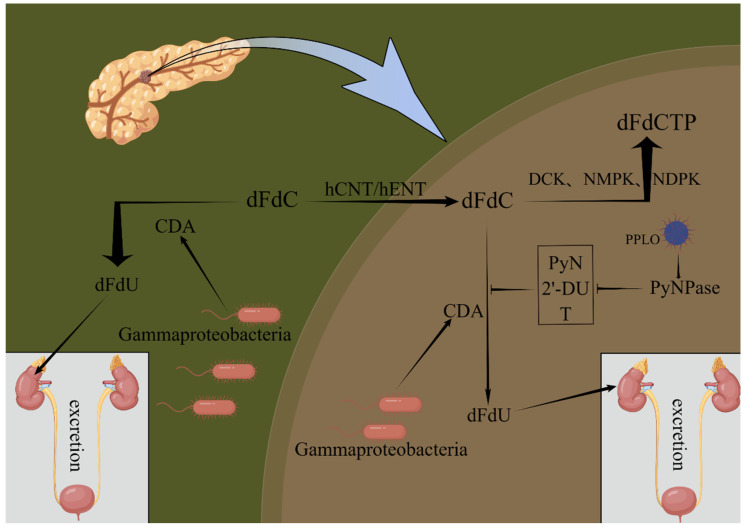
Proteobacteria and *Mycoplasma* affect chemotherapy drugs. Proteobacteria could metabolize gemcitabine by expressing CDA. In addition, *Mycoplasma* can also indirectly enhance the deamination of chemotherapy drugs by removing natural pyrimidine nucleoside, 2′-deoxyuridine, and thymidine. PPLO: patent-pending laboratory organism (mycoplasma); dFdC: difluorodeoxycytidine; dFdU: difluorodeoxyuridine; dFdCTP: difluorodeoxycytidine triphosphate; hCNT/hENT: human concentrative/equilibrative nucleoside transporter; DCK: deoxycytidine kinase; NMPK: nucleoside monophosphate kinase; NDPK: nucleoside diphosphate protein kinase; PyN: pyrimidine nucleoside; 2′-DU: 2′-deoxyuridine; T: thymidine. By Figdraw, www.figdraw.com (accessed on 5 March 2023).

**Figure 2 biology-12-01151-f002:**
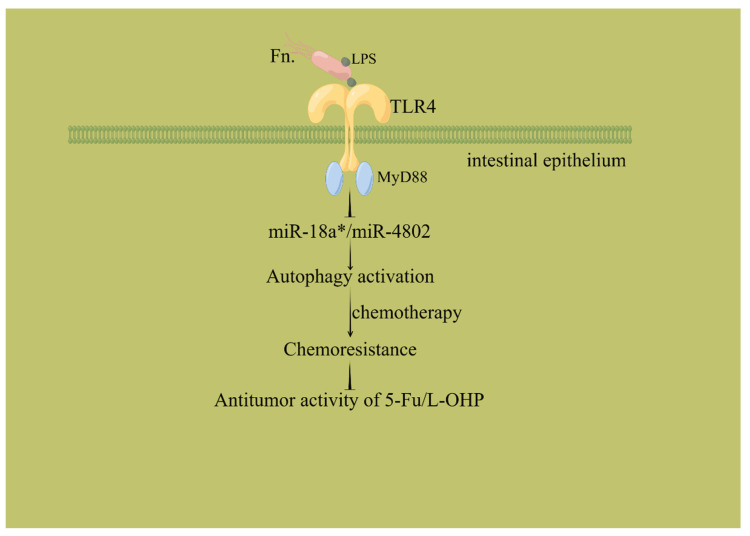
*Fusobacterium* affects chemotherapy drugs. Through Toll-like receptor 4 (TLR4) and myeloid differentiation factor 88 (Myd88) of the TLR signaling pathway, *Fusobacterium nucleatum* induces the selective loss of two autophagy-related microRNAs (microRNA-18a [miR-18a] and microRNA-4802 [miR-4802MYD88]), which activate autophagy, thereby reducing the antitumor activity of oxaliplatin and 5-fluorouracil. Fn.: *Fusobacterium nucleatum*; LPS: lipopolysaccharide; miR-18a*: ‘*’ represents the by-product with a relatively low content; L-OHP: Oxaliplatin. By Figdraw, www.figdraw.com (accessed on 5 March 2023).

**Figure 3 biology-12-01151-f003:**
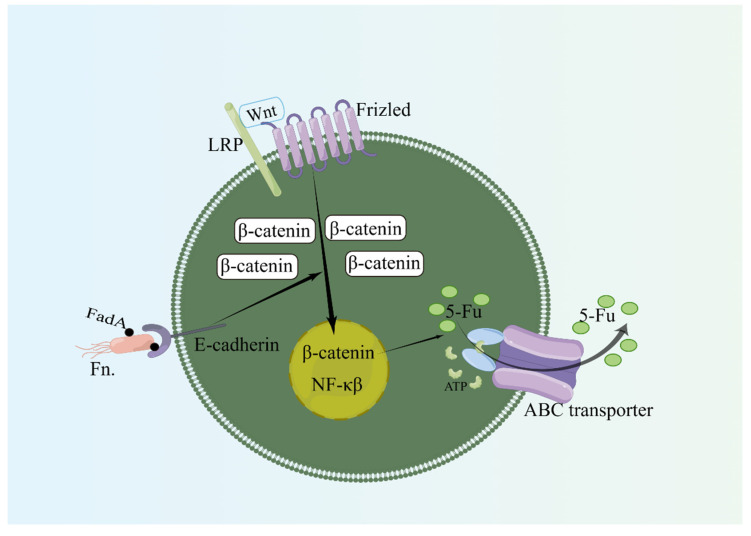
*Fusobacterium* affects chemotherapy drugs. Co-culture with *Fusobacterium nucleatum* can enhance the vitality of cancer cells, reduce cell apoptosis, antagonize 5-FU, and enhance chemotherapy resistance and cancer cell proliferation. This is related to the overactivation of the Wnt/β-catenin signaling pathway, which upregulates the ABC transporters (MRP1 and P-gp). LRP: lipoprotein receptor-related protein. By Figdraw, www.figdraw.com (accessed on 4 March 2023).

**Figure 4 biology-12-01151-f004:**
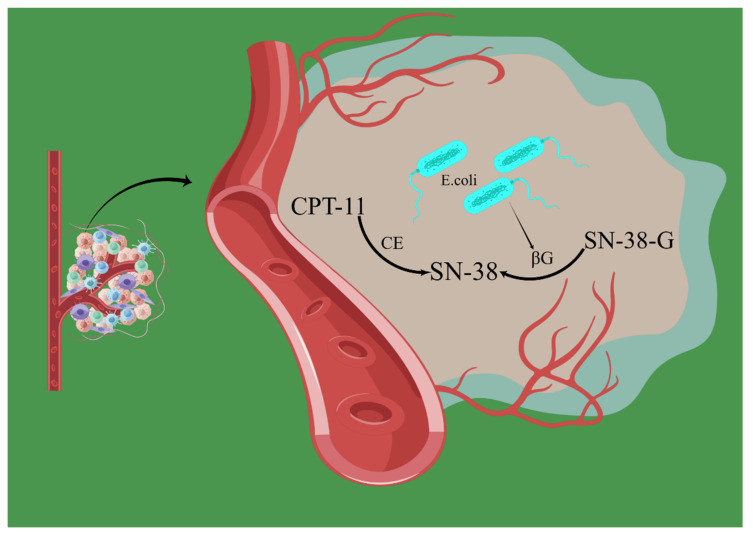
*E. coli* affects chemotherapy drugs. The β-glucuronide produced by *Escherichia coli* can convert inactive glucuronide (Sn-38-G) into active SN-38, inhibit β-glucuronide, and reduce irinotecan activity. *E. coli*: *Escherichia coli*; βG: β-glucuronide; CE: carboxyl esterase; CPT: irinotecan. By Figdraw, www.figdraw.com (accessed on 6 March 2023).

**Figure 5 biology-12-01151-f005:**
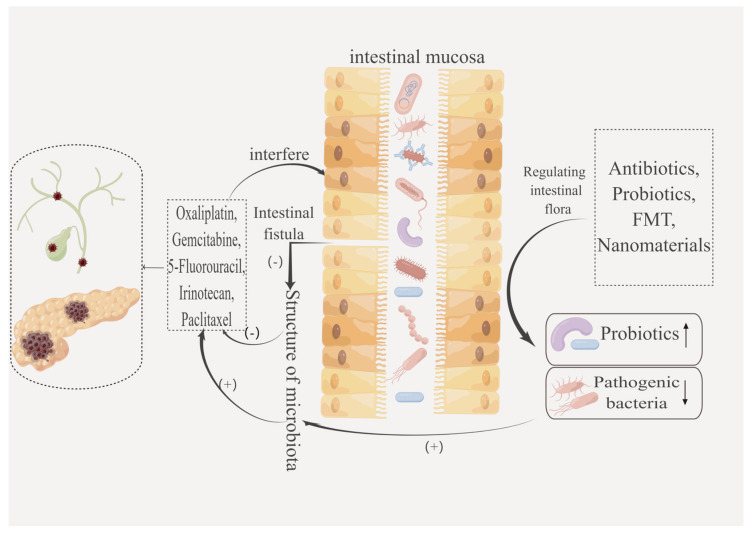
Targeted regulation of intestinal flora by antibiotics, probiotics, and FMT can reduce chemotherapy resistance and improve the efficacy of chemotherapy. ↑: the content increased; ↓: the content decreased. By Figdraw, “www.figdraw.com (accessed on 19 August 2023)”.

**Table 1 biology-12-01151-t001:** Classification of commonly used chemotherapy drugs for biliary pancreatic cancer.

Classification	Representative Drugs	Mechanism of Action	Mechanism of Drug Resistance	Drug-Resistant Flora	Literature
Alkylating agents					
Platinum analogs	Oxaliplatin, cisplatin	Produces unstable alkyl R-CH2 +, which reacts with nucleophilic centers on proteins and nucleic acids, inhibiting DNA replication and transcription.	Increased expression of drug transporters and autophagy regulators.	*Fusobacterium*	[13,21,24]
Antimetabolites					
Cytidine analogs	Gemcitabine	Direct incorporation into DNA and inhibition of DNA polymerase.	Change of drug metabolism, tumor stem cell function.	*Proteus, Bacteroides*, *Mycoplasma*	[13,21,24,25,26,27,28,29]
Pyrimidine analogue	5-fluorouracil, capecitabine	It forms stable covalent complexes with thymidine synthetase and interferes with DNA synthesis and repair.	Increased expression of drug transporters and autophagy regulators.	*Fusobacterium*	[13,21,23,30,31]
Antibacterial drug					
Topoisomerase I inhibitors	Irinotecan	Ternary complexes are formed by preventing topoisomerase I from being released from the cleavable complex to prevent degradation.	Change of drug metabolism, anti-apoptotic.	*Proteus*, *Enterobacterium*	[21,24,30,32,33,34]
Taxanes	Paclitaxel	Promoting microtubule polymerization inhibits depolymerization, interferes with microtubule assembly, and leads to abnormal cell function and replication destruction, leading to cell apoptosis.	Change of drug metabolism, tumor stem cell function.	*Proteus*, Firmicutes, *Bacteroides*	[21,35,36,37]

## Data Availability

This is a review article and does not contain any research data.

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
