# Peer review of "Intestinal Flora in Chemotherapy Resistance of Biliary Pancreatic Cancer"

_biology, 2023, doi:10.3390/biology12081151_

Round 1
Reviewer 1 Report
Clarify the research objective: The abstract and simple summary provide a clear overview of the importance of studying the role of gut microbiota in chemotherapy resistance of biliary pancreatic malignancies. However, it would be beneficial to explicitly state the specific objective of the review at the beginning of the paper. For example, you could clearly state that the aim is to explore the relationship between gut microbiota and chemotherapy resistance and propose potential treatment directions based on this knowledge.
Provide background information: While the abstract briefly mentions that biliary pancreatic malignancy has a high degree of malignancy and poor prognosis, it lacks more specific background information about the current state of chemotherapy treatment for these malignancies. It would be helpful to include a brief background section in the introduction that presents relevant statistics, common chemotherapy regimens used, and their limitations. This will provide readers with a better context for understanding the significance of studying gut microbiota in relation to chemotherapy resistance.
Highlight the significance of the study: In the conclusion, emphasize the significance of studying the role of intestinal flora in chemotherapy resistance for biliary pancreatic malignancies. Explain how this research can potentially lead to improved treatment strategies, better patient outcomes, and a reduced burden of recurrence.
Future research prospects: Provide specific recommendations for future research directions. Highlight areas where further investigation is needed to fully understand the complex interaction between gut microbiota and chemotherapy response in biliary pancreatic malignancies. Consider mentioning potential research approaches, such as preclinical models, clinical trials, or large-scale cohort studies.
In the simple summary, consider rephrasing "chemoresistance leads to poor efficacy" to "chemoresistance reduces treatment effectiveness" for clearer phrasing.
Overall, the quality of the English language in the simple summary and abstract is quite good. The sentences are well-structured and coherent, and the key points are clearly conveyed.
Author Response
请参阅附件。

Reviewer 2 Report
In their manuscript “Intestinal flora in chemotherapy resistance of biliary pancreatic cancer,” Bai et al. intended to critically evaluate and analyse the current state of research on intestinal flora's role in chemotherapy resistance seen in biliary pancreatic cancers. The objective is to provide novel insights and ideas that might potentially contribute to developing more effective treatment strategies for this condition. Following an analytical review, the authors described what they found:
1. The gut microbiota may control chemotherapy efficiency by regulating local immune response and inflammation around tumors. Intestinal flora may modulate cancer autophagy via signaling pathways, affecting chemotherapy treatment resistance and suggesting that gut bacteria may contribute to chemotherapy-resistant tumors.
2. Gut microbiota is involved in the resistance mechanism of different conventional therapy strategies.
3. The modulation of the gut microbiota using antibiotics, probiotics, fecal microbiota transplantation, or nanotechnology has the potential to mitigate chemotherapy resistance and augment the therapeutic efficacy of chemotherapy drugs in combating tumors. This constitutes a large, comprehensive, and broadly rational body of work that is appreciable. However, a number of minor concerns can be resolved to improve the quality of the manuscript, listed below.
1. It Is better to provide a brief discussion on if there is a change in intestinal flora on the basis of the disease progression, like inflammation to advance malignancy.
2. As biliary pancreatic cancer constitutes of different categories, is the intestinal flora contents remain the same among different types of biliary pancreatic cancer?
3. Various factors, such as changes in dietary patterns, adoption of certain diets, or the metabolites produced by the gut microbiota, can potentially modify the makeup of the intestinal flora, disrupt intestinal permeability, and impair the integrity of the intestinal barrier. Put some light on it, please.
4. There is room for grammatical improvement.
There is room for grammatical improvement.
Reviewer 3 Report
The manuscript entitled "Intestinal flora in chemotherapy resistance of biliary pancreatic cancer" reviewed the role of gut microbiota in chemoresistance of malignant tumors of the biliary pancreatic system. The review is well written and cover most of the topics in the field except few of the new findings. The review completely missed the fungal mycobiome in PDAC an emerging field in biliary pancreatic cancer which need to be included.
Alam, A., et al. (2022). Fungal mycobiome drives IL-33 secretion and type 2 immunity in pancreatic cancer. Cancer Cell 40, 153–167.e11. e111. https://doi.org/10.1016/j.ccell.2022. 01.003.
Dohlman et al., 2022, Cell 185, 3807–3822 September 29, 2022 ª 2022 Elsevier Inc. https://doi.org/10.1016/j.cell.2022.09.015
Narunsky-Haziza L, et al. Pan-cancer analyses reveal cancer-type-specific fungal ecologies and bacteriome interactions. Cell. 2022 Sep 29;185(20):3789-3806.e17. doi: 10.1016/j.cell.2022.09.005. PMID: 36179670; PMCID: PMC9567272.
Also, introduction need to be re-written with addition of survival data, pictures need to be improved with significant mechanistic insight and future directions.
Sentence framing need to be improved and redundant sentences need to be removed through out the manuscript.
Round 2
Reviewer 3 Report
Manuscript is improved significantly and all the suggestions are included in the article.
Author Response
According to your suggestion, I modified the article and got your approval. Thank you very much for your guidance.